# Human Preferences for Dogs and Cats in China: The Current Situation and Influencing Factors of Watching Online Videos and Pet Ownership

**DOI:** 10.3390/ani14233458

**Published:** 2024-11-29

**Authors:** Xu Zhang, Yuansi He, Shuai Yang, Daiping Wang

**Affiliations:** 1Key Laboratory of Animal Ecology and Conservation Biology, Institute of Zoology, Chinese Academy of Sciences, 1 Beichen West Road, Chaoyang, Beijing 100101, China; zhangxu@ioz.ac.cn (X.Z.); heyuansi22@ioz.ac.cn (Y.H.); yangshuai@ioz.ac.cn (S.Y.); 2University of Chinese Academy of Sciences, Beijing 100049, China

**Keywords:** human preferences, pet(s), dogs, cats, heritability

## Abstract

Dogs and cats are among the most popular pets, but why humans are so attached to them remains unclear. In this study, we investigate the current state of human preferences for dog and cat videos in China and the potential influential factors behind it. Our findings reveal that current human preferences for dog and cat videos are relatively higher than for most other interests, video plays ranking among the top three out of fifteen interests. Women, young people, and high-income groups are more likely to prefer dog and cat videos. We also find that dog and cat ownership is significantly associated with parents’ pet ownership of dogs and cats. Our study provides new insights into why humans are so attached to dogs and cats and provides a basis for future research on the co-evolution between humans and pets.

## 1. Introduction

Dogs and cats are among the most important pets for humans, providing companionship, emotional support, and other benefits [1,2,3]. Despite ongoing debates about the time and place of dogs’ origin [4,5,6], it is widely accepted that the dog was probably the first domesticated animal [7]. They have been companions to humans for at least 15,000 years, according to the archaeological record, and even longer, according to genome analysis [8,9,10]. Cats have a relatively shorter history of domestication, with evidence supporting a timeline dating back to 9500 years ago [11,12]. Still, both dogs and cats hold an important place in humans’ hearts and lives. According to the latest survey of the American Pet Products Association (APPA), approximately 49% of households in the United States own dogs, and 35% of households own cats in 2023–2024, far exceeding the ownership of ornamental fish, which ranks third with the proportion of 10%. Humans are so attached to dogs and cats that they are willing to lavish much affection, time, and money on their pets, even though this investment does not directly benefit their fitness [13,14,15]. Studies indicate that dog and cat owners view their pets as family members [16,17,18]. To some extent, dogs and cats can even occupy social positions similar to those of partners [19,20] and children [21] in the home.

Why humans are so attached to dogs and cats remains a long studied question, and it was one of the 125 scientific questions jointly published by Shanghai Jiao Tong University (SJTU) and the journal of ‘Science’ (Biology: Why do humans get so attached to dogs and cats?) in 2021 “https://www.science.org/content/resource/125-questions-exploration-and-discovery (accessed on 1 July 2023)”. Previous studies have offered some explanations for this question. Dogs and cats were likely first domesticated for their keen sense of smell [22] and strong territorial instincts [23], which made them valuable for functional roles (e.g., as hunters and protectors) in hunting, herding, and guarding food from rodents [24,25]. Even in modern life, preferences for dogs and cats were often driven by practical purposes. In some remote areas, preferences for dogs and cats remain influenced by their utilitarian roles [26,27], such as herding and guarding. On the other hand, some people keep dogs or cats since they can offer social, physical, and psychological support. Dogs and cats may provide similar companionship as family members [19,20,21,28]. In addition, numerous studies have shown that they can relieve people’s mental stress, improve emotional well-being, and provide therapeutic support [15,29,30,31,32,33,34]. Studies show that interacting with pets can significantly increase owners’ activity levels, thereby improving physical health [35,36]. From a mental health perspective, the psychological support and comfort provided by dogs and cats is a key reason people choose them [37,38,39,40]. Human preferences for dogs and cats may also be influenced by genetic variations. A study on twins showed that the heritability of dog ownership is high in both genders (57% for women and 51% for men) [41]. Apart from these, there are other factors that may affect human preferences for dogs and cats, such as gender, age, or individual income. For instance, it is widely accepted that women tend to be more compassionate and exhibit stronger attachment behaviors, affect, and caregiving than men [42,43], younger adults have higher cognitive empathy than older adults [44,45], which may make them more attached to dogs and cats. In addition, age and wealth may determine how much energy, time, and money a person can devote to dogs and cats and thus affect the degree of preferences [14]. Additionally, from a sociocultural perspective, keeping dogs and cats is considered a positive behavior in some religions or cultures [46]. Influences from celebrities and media often lead their followers to develop interests in specific pets and mimic their behaviors, such as social posts by pet influencers that can affect pet owners’ purchasing decisions [47,48]. This not only strengthens the emotional bond between humans and pets but also positively impacts family and community.

These potential factors indicate a complex and multidimensional co-evolution between humans and pets (dogs and/or cats). Humans tend to choose pets that meet their survival, work, psychological, and social needs, thereby enhancing their fitness. Correspondingly, dogs and cats gain opportunities for survival and reproduction by being selected by humans, which further reinforces and continues the evolution of traits preferred by humans. Animal behavior theories further explain how pet behaviors adapt to meet human psychological needs, such as their responsiveness to human emotions and adaptability to various environments [49,50,51,52].

Though many studies have attempted to explain why humans get so attached to dogs and cats, most of them mainly focus on one single factor and lack generality. To better answer this question, a more comprehensive and systematic investigation is required, especially its current situation and potential influencing factors. Given that humans have developed a variety of preferences, such as games and music, it is necessary to determine to what extent humans currently prefer dogs and cats compared to other hobbies and interests, such as music and games, and how different factors influence that preference. In fact, most YouTube users actively seek out and watch content that aligns with their interests [53]. Research on YouTube cat videos has indicated a significant correlation between people’s affinity (preference) for cats and the views of these videos [54]. Therefore, we will analyze human preferences for dogs and cats based on their preferences for dog and cat videos, as well as their behaviors related to pet ownership. Here, we used data from online video platforms and anonymous questionnaires to obtain a more comprehensive and systematic overview of the human preferences for dogs and cats. We first investigated how much more humans prefer videos about dogs and cats compared to videos on other topics. Then, we calculated the relationship between people’s preferences for dogs and cats and those of their parents based on the information about dog and cat ownership from the questionnaire respondents and their family members. Meanwhile, we investigated the main reasons why they chose or planned to have dogs or cats. Finally, we verified the potential factors (i.e., age, gender, individual income, and psychological factors) that may have an effect and the extent of their impact by analyzing the basic information of video audiences and questionnaire respondents. Specifically, we posited that young people, women, and those with high incomes were likelier to prefer dog and cat videos than older people, men, and those with lower incomes.

## 2. Materials and Methods

### 2.1. Data Collection

#### 2.1.1. Bilibili Videos

To get an overview of human preferences for dog and cat videos, we chose Bilibili to collect video data. It is one of the leading streaming platforms in China, containing 326 million users up to 2022. It provides videos on various topics such as anime, music, games, fashion, movies, and pets, and users are allowed to watch, like, and comment on these videos. Meanwhile, the platform classified different videos into different channels based on their subject matter, which makes it easy for us to get the audience’s preferences for different types of videos. Bilibili has selected 96 channels as ‘popular channels’, containing almost all of the most popular videos. To avoid over-categorization, we reclassified the 96 popular channels into 15 categories according to the channel’s theme (Appendix A). For example, three pet-related channels (“Cute pets”, “Dogs”, and “Cats”) are grouped into the “Pets” category. For each channel, we defined videos with more than one million views as “popular videos”. We then counted (a) the number of popular videos, (b) the number of plays of popular videos, and (c) the number of likes of popular videos for each category from 2009, when the platform was established, to 2021, using them as indicators of human preferences for different types of videos.

#### 2.1.2. Online Questionnaires

To understand the influence of genetic and other factors on human preferences for dogs and cats, we designed an online questionnaire “https://wj.qq.com/s2/13778005/f591/ (accessed on 1 December 2023)” using Tencent Questionnaire “https://wj.qq.com/ (accessed on 1 December 2023)” and distributed it to the user base of the platform, as well as WeChat and QQ, the major communication platforms in China in January 2024. Detailed questions are listed in Appendix A. To avoid influencing participants’ responses, we did not disclose specific hypotheses, but they were informed of the general purpose of the study. We ensured that the data from anonymous users would be handled appropriately. The questionnaire was divided into four main sections to capture details about the human–pet relationship comprehensively. As shown in Appendix A, Questions 1 to 5 were intended to analyze whether people’s ownership of dogs and cats was significantly influenced by their parents. Questions 6 to 10 were aimed at capturing the main reasons why respondents chose to own dogs or cats. For Questions 11 to 20, Questions 13, 16, and 20 were formulated to assess whether dogs and cats could partially fulfil the role of close family members, or even substitute for them in certain context, and the other questions were used to determine which respondents needed to answer Questions 13, 16, and 20. Questions 21 to 25 were designed to gather basic demographic information from respondents, allowing us to assess the sample’s representativeness.

To ensure data quality, we excluded responses with obvious errors (incomplete answers or unreal answers) during data collection, resulting in 500 valid questionnaires. Among the respondents, 180 were men (36%) and 320 were women (64%). The age of the respondents ranged from 13 to 59 years, with an average age of 25.32 years (SE = 0.32). The respondents were from 30 provincial-level regions across China (out of a total of 34), with 140 (28%) from rural areas and 360 (72%) from urban areas. The average monthly income of respondents was CNY 5884 (SE = 506). The sample reflected the preferences for dogs and cats among Chinese internet users, mainly young people.

#### 2.1.3. Douyin (TikTok) Videos

Douyin (TikTok) is another mainstream short-video platform in China. It is easier to access and reaches a larger number of users. To collect data on dog and cat videos from Douyin, we used a data analysis platform called Feigua “https://dy.feigua.cn (accessed on 1 July 2023)”. This platform will rank videos each month according to the number of comments and provide basic information such as the number of plays of videos and comments. Also, it can offer information such as the gender (man or woman), age distribution (6 age groups: 0–17, 18–23, 24–30, 31–40, 41–50, and over 50 years old), and the region (the top 10 provinces in China in terms of number of commenters) of the video audience based on the sample of the latest commenters. To analyze how different factors affect people’s preferences for dog and cat videos, we collected the above information from the top 100 pet videos every month from August 2022 to July 2023. Since pet videos include animals other than dogs and cats, we ended up with 1006 dog and cat videos out of 1200.

### 2.2. Data Analyses

#### 2.2.1. The Current Situation of Human Preferences for Dog and Cat Videos

We obtained the current human preferences for dog and cat videos from the dataset collected from Bilibili. Different categories were ranked according to their median number of plays, likes, and popular videos per category per year so that we could see how much people liked dog and cat videos relative to other types of videos. We removed overlapping videos across different channels of the same type to avoid multiple counting. Moreover, we used linear regression to analyze how people’s preferences for different categories of videos change over time.

#### 2.2.2. The Parental Influence on Dog and Cat Ownership

For the 500 valid questionnaires collected, we estimated the parental influence on dog and cat ownership by Spearman’s rank correlations (spearman’s ρ) [55] according to whether the respondents and their parents and grandparents own or have owned dogs or cats. We interpreted the transition from not owning pets to owning only one type, and eventually to owning both dogs and cats, as a form of stepwise behavior. Therefore, we treat the behavior of owning dogs and/or cats as an ordinal variable. Specifically, we used “0” to represent family members without pets, “1” for those who own or have owned dog(s) or cat(s), “2” for those who own or have owned dog(s) and cat(s). Then, we calculated the parental influence of dog and/or cat ownership by calculating the mean of Spearman’s rank correlation coefficients of dog and/or cat ownership between offspring and their parents, parents and paternal grandparents, and parents and maternal grandparents, using Fisher’s r-to-z transformation to obtain the average. Additionally, we used the MCMCglmm package [56] in R 4.2.2 [57] (i.e., a typical quantitative genetics method for estimating heritability in Animal Models, which incorporated pedigree information) to estimate the potential heritability of dog and cat ownership behavior as a reference.

#### 2.2.3. The Reasons for Owning Dogs and/or Cats

Among the 472 respondents with experience or a plan to adopt dogs or cats, we listed five main reasons for having dogs or cats and asked them if they agreed. These options were (1) for health improvement, (2) because of religious reasons or cultural traditions, (3) to gain emotional value (or compensation), (4) for practical uses like guarding or catching rats (treating them as tools), and (5) to follow the trend (i.e., people around them have pets). By calculating the proportion of yes or no for each option, we can know whether people prefer dogs and cats because of their utility attributes, or to provide psychological comfort.

#### 2.2.4. Factors That Influence the Preferences for Dog and Cat Videos

We used the dataset of Douyin (TikTok) videos to investigate how potential factors influence human preferences for dog and cat videos. Analyses were done by fitting linear mixed-effects models using the lme4 package [58] in R 4.2.2 [57]. We used the number of comments on videos as the response variable to represent the audience’s preferences. The identity of the videos (Video_ID) and the region where the publisher was located (Region) were taken as random effects, while the type of the videos (two levels: dog or cat), the gender (two levels: man or woman), the age (6 levels: 0–17, 18–23, 24–30, 31–40, 41–50, or >50 years old), and the economic level (GDP, a continuous variable) of the audience were taken as fixed effects.

## 3. Results

### 3.1. The Current Situation of Human Preferences for Dog and Cat Videos

According to our statistics, humans prefered videos that include dogs and cats to most other subjects (Figure 1). Regarding the median of plays for popular videos, the Pet category ranked third out of 15, while it ranked eighth in terms of likes and number of popular videos. Furthermore, we found that the growth trend of these three indicators in the Pet category ranked eighth out of 15 categories, indicating the growing trend in human preferences for dog and cat videos is relatively moderate across all categories (Appendix A).

### 3.2. The Parental Influence on Dog and Cat Ownership

In this study, the Spearman’s rank correlation coefficient (spearman’s ρ) for the relationship between dogs and cats ownership and parental ownership of dogs and cats was 0.43 (95% CI: 0.38–0.47). Specifically, the Spearman’s rank correlation coefficients of three groups, Parents–Offspring, Paternal grandparents–Parents, and Maternal grandparents–Parents, were 0.47 (95% CI: 0.38–0.55), 0.41 (95% CI: 0.33–0.50), and 0.40 (95% CI: 0.30–0.49), respectively (Figure 2 and Table 1). The potential heritability of dog and cat ownership behavior was estimated to be 0.35 (95% CI: 0.29–0.39) using the MCMCglmm package in R.

### 3.3. The Reasons for Owning Dogs and/or Cats

As for the reasons for keeping a dog (dogs) or a cat (cats), 85.8% of the respondents agreed that keeping dogs and cats had an emotional value, 31.6% considered that it is good for health, 41.3% treated dogs and cats as tools, 25.2% of the respondents kept dogs and cats to follow the popular trend, and only 6.1% chose to do so for religious or cultural reasons (Figure 3). Furthermore, among those who acknowledged owning a dog or cat has emotional value, most respondents agreed that dogs and cats can compensate for the emotional void from the absence or loss of family members (Appendix A).

### 3.4. Factors That Influence the Preferences for Dog and Cat Videos

We found that videos about dogs received more comments than videos about cats (t = 3.7, *p* < 0.001, df = 1004, Table 2 and Figure 4). Female viewers commented more than male viewers (t = 137.4, *p* < 0.001, df = 119,672). At the same time, the level of economic development also impacted the audience’s preferences. More commenters came from the region with a higher level of economic development (e.g., Guangdong, t = 7.6, *p* < 0.001, df = 23). Among the viewers who left comments, the proportion of different age groups was also significantly different. In general, young people left more comments than older people.

## 4. Discussion

While previous studies have provided some explanations for why humans are so attached to dogs and cats, a more comprehensive and systematic overview of the human preferences for dogs and cats is required. Using three independent datasets (i.e., Bilibili videos, online questionnaires, and Douyin (Tik Tok) videos), our results suggest that people may have a preference for videos that include dogs and cats over other subjects in China (Figure 1). Whether individuals own cats, dogs, or both is significantly associated with whether their parents do (Figure 2 and Table 1). Gender, age, and economic development impacted the audience’s preferences significantly (Figure 4 and Table 2). Overall, our study highlights the current situation and influencing factors of human preferences for dog and cat videos in China.

It should be noted that the data in this study were basically sourced from online platforms, which might contain biases and errors [59]. Our results might not fully capture the preferences for dog and cat videos among the elderly and those with limited access to the internet [60]. It is also challenging to completely eliminate self-reporting bias that may be present in online surveys [61]. However, under certain conditions, it is still possible to derive accurate conclusions from biased data [62,63]. Also, given that our data is limited to China, the impact of cultural and regional differences should also be considered. However, overall, our findings remain valid and our conclusions can reflect the actual situation in the real world. Our findings offer significant insights into the preferences for dog and cat videos among Chinese internet users, particularly young people. Future research should employ additional methods and approaches to further mitigate biases present in online data, and could involve cross-cultural and regional perspectives to study human preferences for dog and cat in different cultures and areas around the world.

Surveys have compared the proportion of households owning dogs, cats or other types of pets within the overall pet-owning population [64,65,66,67,68]. However, few studies try to compare the level of preferences for dog and cat videos with other preferences in modern life. In this study, using the dataset of online videos (i.e., Bilibili videos), we found that the number of views for both dog and cat videos is relatively high compared to other types of preferences (Figure 1 and Appendix A). This pattern gives us a more comprehensive picture of the current situation of human preferences for dog and cat videos. Moreover, the results also indicated that humans are indeed attached to dogs and cats [69,70]. Since the results here were based on online videos, they might not totally reflect the situation of daily life. For instance, most people might be biased toward games, fun videos, and movies that only exist virtually online. This might underestimate the degree of preferences for dogs and cats. Another limitation that needs to be considered is that people with dogs or cats in daily life might spend less time on the internet (e.g., they have to look after their pets). In this respect, future studies directly investigating the preferences for dogs and cats in daily life are more appealing.

In this study, we found that whether individuals own dogs, cats, or both is significantly linked to whether their parents do (Figure 2), suggesting a potential genetic basis. Our calculated Spearman’s rank correlation coefficient is 0.43 (95% CI: 0.38–0.47). Additionally, we calculated that the potential heritability of dog and cat ownership behavior was 0.35 (95% CI: 0.29–0.39), which is very close to a recent study on monozygotic and dizygotic twins showing that genetic factors account for 37% of the differences in pet play [71]. Similarly, results from Fall et al. indicated that the heritability of dog ownership was 0.51 for men and 0.57 for women among monozygotic and dizygotic twins [41]. These similarities between our study and previous studies indicate that human preferences for dogs and cats are potentially influenced by genetic variations [71,72]. In conclusion, we found that the behavior of owning dogs and cats is significantly associated with parental behavior, indicating the possible influence of genetic factors. Since our results primarily focus on correlation analysis, they cannot clearly distinguish the specific influences of genetics and the environment on pet ownership behavior. Therefore, these findings should be interpreted with caution, and future research could further clarify the relative contributions of these factors through methods such as twin or adoption studies.

Furthermore, previous studies showed that functional roles and companionship are the two main reasons for having dogs, cats, or both [73,74,75,76,77]. Yet, these studies did not compare the selection frequencies of these reasons simultaneously or within a study. In our study, we have listed five main reasons at the same time and evaluated their selection frequencies. We found that the largest number of respondents kept dogs and cats to obtain emotional support (*N* = 405), followed by treating dogs and cats as tools (*N* = 195), keeping them for health reasons (*N* = 149), keeping them to follow the popular trend (*N* = 119), and keeping them for religious or cultural reasons (*N* = 29) (Figure 3). These new results offered new insights into the research on human attachment to dogs and cats. In the initial stages of the relationship between humans and pets, the primary consideration for most people was the functional roles of dogs and cats. Historically, dogs assisted humans with hunting, guarding, and protection [78,79,80,81,82], while cats helped humans by protecting food supplies [83,84]. However, as societal productivity developed, the need for emotional companionship gradually became the main reason for most people to keep dogs and cats [85,86]. From a social structure perspective, the proportion of single-person households is at an unprecedented high in Western countries [87], with similar trends emerging in China [88,89]. Given that living alone could significantly enhance feelings of loneliness [90], and social isolation, which could have severe negative impacts on mental health [91,92,93], owning dogs and cats as companions can substantially alleviate loneliness for individuals [94,95]. Pets not only help their owners to form new relationships as a catalyst [96], but also substitute for human-provided social support [97]. Additionally, attachment theory in psychology was applicable to explain human attachment to companion animals as well [19]. Interacting with dogs and cats could generate positive emotions [98], and the level of intimacy with companion animals could also influence people’s sense of well-being [99]. Our questionnaire results (i.e., obtaining emotional value) indicated that people also consciously choose to own dogs and cats to achieve a better psychological and mental state (Figure 3). In short, this shift from most people treating dogs and cats as tools to valuing their role in psychological and mental health highlights a dynamic process of human preferences for dogs and cats, reflecting an evolutionary change from practical utility to emotional appreciation, which is an interesting and valuable research topic and deserves further study.

In addition to genetic variations, other potential factors may affect human preferences for dog and cat videos, such as gender, age, or individual income. Generally, we found that women, younger people, and those with higher incomes had stronger preferences for dog and cat videos compared to men, older people, and people with lower incomes (Table 2 and Figure 4). Additionally, our findings reveal that humans show a higher level of interest in dogs compared to cats (Table 2 and Figure 4a), which corresponds with the status of dogs being the most popular pets in Sydney (33.4% of households owned dogs, and 22.5% owned cats) [65]. In previous studies, how different factors affect human preferences for dogs and cats has not yet come to a unified definition. Our findings (Table 2 and Figure 4b,c) align with a recent study reported by Fraser showing that younger people and women are more likely to be pet owners in New Zealand [100]. Similarly, some studies support the notion that women have stronger preferences [101,102,103], while a few suggest that men have stronger preferences [104]. These discrepancies may be attributed to regional differences, with varying cultural traditions potentially influencing the results. When it comes to individual income, we found that regional GDP was positively correlated with the number of comments. Higher income levels likely make it more feasible to afford the costs of owning dogs and cats. People living in a relatively more developed area may also find it easier at work and have more leisure time to keep a dog or cat [105].

In this study, the result that younger individuals, women, and people from more economically developed regions in China show stronger preferences for dog and cat videos may be related to the progress of urbanization, cultural shifts, and economic development in the country. In the past two decades, China’s urbanization rate has increased from 40.5% in 2003 to 66.2% in 2023 (data from the Chinese government), which has led to a concentration of many single young adults in major cities [106,107]. The emotional void caused by being away from parents or partners (or not having a partner) in their hometowns may compel young people to choose to view dog and cat videos, or own dogs and cats. Our results indicate that over 67.1% (*N* = 173) and 72.6% (*N* = 379) of respondents deem that owning dogs and cats, respectively, can somewhat alleviate the loneliness caused by the absence of parents or partners (Appendix A). Due to traditional Chinese marriage norms, which expect men to have a better status than women, multiple studies [108,109,110] and media reports [111] indicate that urban single elite women are having increasing difficulty finding suitable partners. Consequently, single women in cities are more likely to develop a preference for dog and cat videos due to emotional or social needs. Additionally, the rapid shift from agricultural culture to industrial culture in China [112], along with the introduction of the foreign concept of animal welfare in the late 20th century [113], may have made young people who grew up in this period more likely to prefer watching online dog and cat videos. The role of economic development should not be underestimated either. As Chinese women gain economic independence [114], more and more women are likely to have the opportunity and time to use electronic devices to watch online dog and cat videos. In conclusion, the social, cultural, and economic transformations brought about by China’s recent development may have considerable influence Chinese preferences for dogs and cats. Notably, each factor that affects the degree of human preferences for dog and cat videos might not be independent of each other, and further research may take into account their potential combined effects to derive more comprehensive conclusions.

In this study, we did not consider the morphological or psychological traits of the animals, such as appearance, age and personality, because they are difficult to quantify from the videos. Previous studies have shown that the traits of dogs and cats can also influence human preferences for them. For instance, several articles indicated that the appearance of dogs is the main factor that highly influences people’s decisions to buy or adopt them [115,116,117]. Humans may choose some features that are thought to be associated with an infantile aesthetic, such as bigger eyes and a larger space between the eyes [118]. Also, dogs that can enhance paedomorphism (change the eye size and height by raising the inner brow) through greater facial flexibility are found to be more desirable to humans [119]. Additionally, the animal’s age also potentially affects people’s decisions on whether or not to have a dog. According to the research by Brown [120], the length of a dog’s stay in a shelter increases with its age, which means the older the dog is, the less likely it is to be adopted. Similarly, studies conducted in Australia and Italy find that the public tends to acquire dogs as puppies [121,122]. This tendency may be associated with the preference for infantile-like features, the opinion that a puppy can be trained to acquire good habits, or just because the owner wants to experience the “puppy stage” of a dog’s life [117]. Apart from this, researchers also find that how animals behave when interacting with humans can affect human attachment to pets, which we cannot fully study in our work with online videos and questionnaires. A study by Protopopova and Wynne [123] shows that dogs that are willing to respond to potential adopters are more likely to be adopted. Similar studies have also found that a dog’s temperament and its behavior during interactions with people can influence human attitudes toward them [115,122]. In short, future work that considers these aspects (i.e., the traits of pets and their social behavior) in conjunction with the characteristics of the owners (such as personality, gender, age, and economic status) could provide a more thorough understanding of factors that influence human preferences for these pets.

Human preferences for dogs and cats are never one-sided. This interesting question involves a complex interplay of evolving human tastes, the selection of various dog and cat breeds, the dynamic characteristics of these animals, and their impact on people. On the one hand, humans selectively breed dogs and cats according to their own preferences and needs, even if certain traits can lower individual fitness. For example, Scottish Fold cats with osteochondrodysplasia [124] and French Bulldogs, which are more prone to health issues than other breeds [125], are still being chosen by humans. On the other hand, some believe that cats and dogs change their ancestral behaviors to function as social releasers, eliciting human parental care [13] and may thus gain higher fitness. In summary, this is a co-evolutionary process of mutual selection. However, research in this area is still lacking, particularly in co-evolutionary model construction and simulations. Comprehensive mathematical simulations and models can delve deeper into the impact of human behavior on the evolution of dogs and cats. We should emphasize these aspects more, as they will help us understand how humans and dogs and cats have influenced each other throughout their long companionship, and why humans are so attached to these animals.

## 5. Conclusions

In conclusion, this study demonstrated that human preferences for dog and cat videos were relatively higher than for most other interests, such as music and games. These preferences are notably influenced by gender, age, and economic development levels. Women, young people, and high-income groups are more likely to prefer dog and cat videos. Our findings reveal that individuals whose parents owned dogs and/or cats are more likely to own dogs and/or cats, indicating a potential hereditary component. The most common motivation for pet ownership identified in our research is the pursuit of emotional support, highlighting the deep emotional bonds humans form with dogs and cats.

Our findings provide insights into why humans are so attached to dogs and cats, and establish a foundation for future research on the co-evolution of human preferences and pets. In the future, more direct data from daily life could be used to analyze human preferences for dogs and cats, and various research approaches could be employed to investigate the heritability of dog and cat ownership.

## Figures and Tables

**Figure 1 animals-14-03458-f001:**
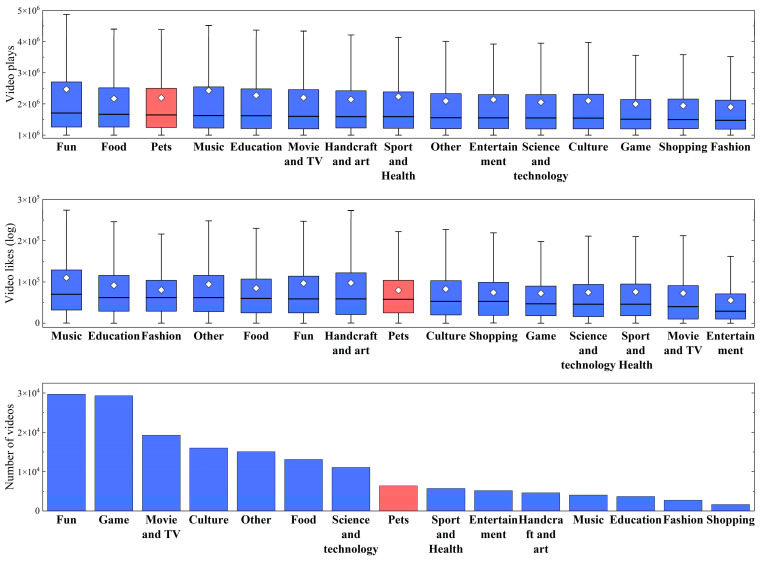
Box plots of the number of plays and likes, and histogram of popular videos of 15 types of human preferences. The diamond shape represents the mean value. The line inside the box shows the median. The box represents the interquartile range (IQR), containing the middle 50% of the data from the 25th percentile (Q1) to the 75th percentile (Q3). The whiskers extend to 1.5 times the interquartile range (IQR) from the 25th percentile (Q1) and the 75th percentile (Q3). Points outside the whiskers are outliers.

**Figure 2 animals-14-03458-f002:**
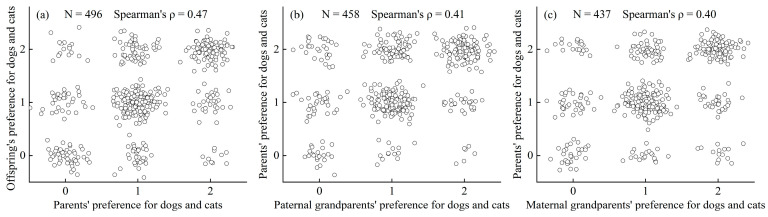
The parental influence on dog or cat ownership: (**a**) Parents–Offspring, (**b**) Paternal grandparents–Parents, (**c**) Maternal grandparents–Parents. Here, “0” means no pets, “1” represents owning dogs or cats, and “2” represents owning both dogs and cats. Spearman’s rank correlations were used to calculate the parental influence on dog or cat ownership.

**Figure 3 animals-14-03458-f003:**
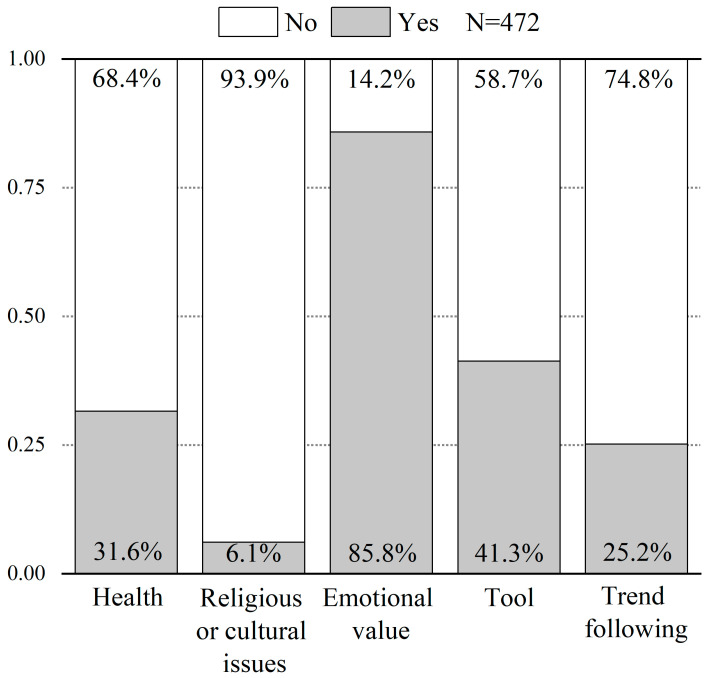
The proportion of respondents (yes or no) who answered five questions about the reasons for having dogs or cats. These five options are: (1) for health improvement, (2) because of religious reasons or cultural traditions, (3) to gain emotional value (or compensation), (4) for practical uses like guarding or catching rats (treating them as tools), and (5) to follow the trend (i.e., people around them have pets).

**Figure 4 animals-14-03458-f004:**
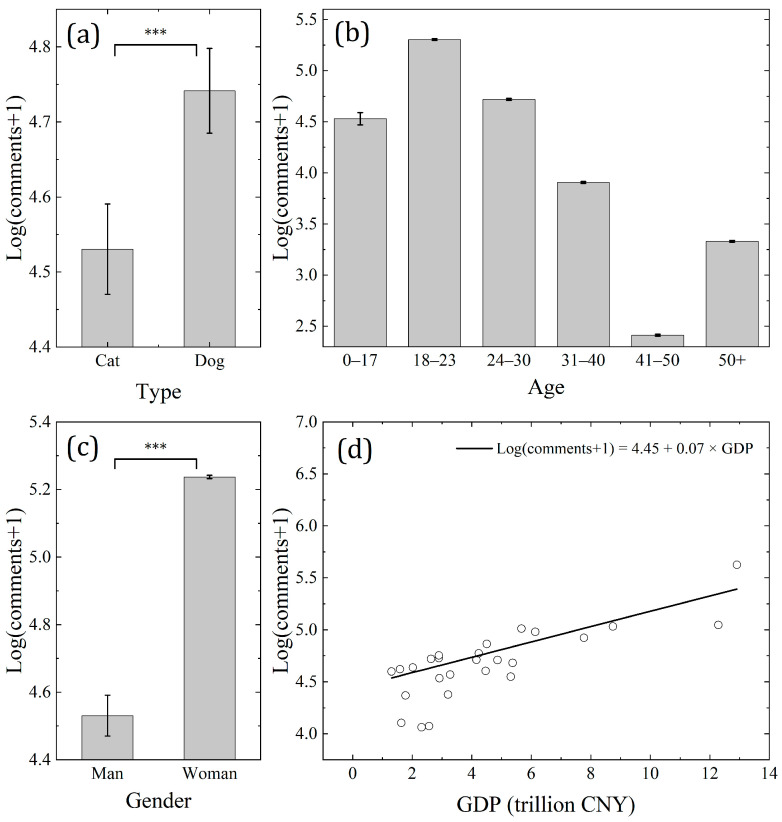
Factors influencing human preferences for dog and cat videos: (**a**) comparison between different types of videos; (**b**) comparison between different age groups; (**c**) comparison between genders; and (**d**) the relationship between the number of comments and the level of regional economic development. The whiskers in Figure 4a–c represent the standard error (SE) values. *** indicates *p* < 0.001 for all comparisons.

**Table 1 animals-14-03458-t001:** The Spearman’s rank correlation for the parental influence on the ownership of dogs and cats.

	Spearman’s ρ	95% CI
Parents vs. Offspring	0.468	0.384–0.552
Paternal Grandparents vs. Parents	0.412	0.326–0.499
Maternal Grandparents vs. Parents	0.398	0.304–0.492

**Table 2 animals-14-03458-t002:** Results of the linear mixed-effects model analysis of factors influencing human preferences for dog and cat videos. We used the number of comments on videos as the response variable to represent the audience’s preferences. The identity of the videos (Video_ID) and the region where the publisher was located (Region) were taken as random effects, while the type of the videos (two levels: dog or cat), the gender (two levels: man or woman), the age (six levels: 0–17, 18–23, 24–30, 31–40, 41–50, or >50 years old), and the economic level (GDP, a continuous variable) of the audience were taken as fixed effects.

Effects	Estimates ± SE	t Value	*p* Value
Random effects			
Video ID *(N* = 1006)	0.794		
Region (*N* = 26)	0.015		
Residual	0.798		
Fixed effects			
Intercept	4.527 ± 0.061		
Woman	0.707 ± 0.005	137.4	<0.001
Dog	0.211 ± 0.057	3.7	<0.001
GDP	0.063 ± 0.008	7.6	<0.001
Age 18–23	0.775 ± 0.009	87.0	<0.001
Age 24–30	0.189 ± 0.009	21.2	<0.001
Age 31–40	−0.625 ± 0.009	−70.1	<0.001
Age 41–50	−2.119 ± 0.009	−237.8	<0.001
Age 50+	−1.200 ± 0.009	−134.7	<0.001

## Data Availability

The data underlying this article are available in the Dryad Digital Repository at https://doi.org/10.5061/dryad.qfttdz0rr.

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
