# Peer review of "Human Preferences for Dogs and Cats in China: The Current Situation and Influencing Factors of Watching Online Videos and Pet Ownership"

_animals, 2024, doi:10.3390/ani14233458_

Round 1
Reviewer 1 Report (Previous Reviewer 2)
Comments and Suggestions for Authors
The authors have done a good job of addressing my concerns.
Author Response
Comments 1:
The authors have done a good job of addressing my concerns.
Response 1:
We sincerely appreciate all the previous suggestions and concerns raised by the reviewer. Addressing these suggestions and issues has significantly enhanced our work.

Reviewer 2 Report (Previous Reviewer 3)
Comments and Suggestions for Authors
Congratulations to the authors, all points have been addressed with success.
Author Response
Comments 1:
Congratulations to the authors, all points have been addressed with success.
Response 1:
We appreciate the reviewer's positive assessment. We believe the review process has greatly enhanced the quality of our work.

Reviewer 3 Report (New Reviewer)
Comments and Suggestions for Authors
Review of Zhang et al.’s Human preferences for dogs and cats in China: the current situation and influencing factors, manuscript id animals-3261832.
Zhang et al. attempt to determine if and why dogs and cats are preferred by residents of China. It is not clear to me that two of their main measures (number of views and number of comments made to videos) are reliable and valid measures of preference for dogs and cats. It also appears that they are trying to interpret random noise in the data. I also question the appropriateness of using linear regression, which is based on bivariate correlation, on ordinally scaled variables. Given these concerns, I am not sure that their discussion follows from their data.
Issues:
The title states “preference for dogs and cats” yet you measured preference for dog and cat videos. You need to justify why the number of views and comments to dog and cat videos are reliable and valid measures of preference for dogs and cats. Perhaps the dog and cat videos are watched more than other types of videos because they are funnier than the other types of videos or cuter than other type of videos or more calming, etc.
Lines 141-143: You state that questions 11 through 20 assess whether dogs and cats can partially fulfill the role of close family members. It is clear how questions 13, 16, and 20 can be used to answer that question. It is not clear how the other questions can be used to determine if dogs and cats can partially fulfill the role of close family members. Please explain.
Why are you using Pearson’s correlation and linear regression with your ordinally scaled data? Pearson’s r requires interval or ratio scaled data. Linear regression makes no sense with ordinally scaled (dummy coded) variables. Your results and discussion need to be rewritten to reflect a proper measure of association.
Line 190: It does not make sense to take the mean of several Pearson’s rs as Pearson’s r is not linear. Consider using an r to z transform, taking the mean of the z-scores, and use a z to r transform.
Lines 191-192: Perhaps this is just a cultural difference. The first question on the questionnaire just asks about “grandparents” and not “paternal grandparents”. The second question on the questionnaire asks about “maternal grandparents”. How did you measure dog and cat ownership of paternal grandparents?
Lines 220-226: To me, you are interpreting random noise in the data. That is, the means and medians do not look like they would be reliably different as they appear to be within each other’s IQRs. Do you have statistical results to support your statement that pet category is third or fourth most popular? If not, lines 289-290 in the discussion will need to be modified.
Lines 323-324: It could also represent an environmental basis – living with a pet while growing up may predispose a person to have a pet during adulthood. You need to explain how your data tease these two factors (nature vs nurture) apart.
Lines 337-341: If I am understanding you correctly, you are reporting the percentage of people who answered “Yes” to question 6 through 10. You are giving the frequencies of each category but that is not the same as “relative importance”. More people might claim that they own a dog or cat for functional purposes (e.g. 100 people) than for emotional value (e.g. 90 people), but the functional purpose people may give an importance rating (which you did not measure) of 2 out of 7 while the emotional people may give an importance rating of 7 out of 7.
Minor issues:
Line 27: Insert a space between the hyphen and 0.51. As written, it looks like the CI runs to negative 0.51 which would imply that the CI contains 0.
Line 50: “study” should be “studied”
Lines 111-112: Close the gap. The materials and methods section should not start on a separate page.
Line 149: Include the unit (e.g. years) with ages.
Line 166: The sentence starting “Furthermore, we obtained…” does not belong in section 2.1.3 Douyin (TikTok) videos. Please put it in a separate section.
Line 208: “datas et” should be “data set”
Figure 3: I recommend that you make the filled part of each bar lighter in color or make the value labels within each bar white. Either way you will increase the contrast of the value labels and bars making the graph more legible.
Line 266: “Female” and “male” would be more appropriate than “Woman” and “man”.
Caption for figure 4: Indicate what the whiskers represent in the figures 4a-4c.
Line 291: “owning” should be “own” and “are” should be “is”
Line 371: Lines 42-43 state 49% and 35% of households in the USA own dogs / cats. Line 371 states 33.4% and 22.5% of households own dogs / cats. I am guessing that the difference is due to the region (USA vs not specified). It would be nice if you made explicit the difference in these percentages.
Line 381: “data from Wiki” is not an appropriate citation or source.
Line 384: Consider deleting “real”. “Life” is “real” by definition.
Line 413: Consider “articles” instead of “research”.
Author Response
Comments 1:
Review of Zhang et al.’s Human preferences for dogs and cats in China: the current situation and influencing factors, manuscript id animals-3261832.
Zhang et al. attempt to determine if and why dogs and cats are preferred by residents of China. It is not clear to me that two of their main measures (number of views and number of comments made to videos) are reliable and valid measures of preference for dogs and cats. It also appears that they are trying to interpret random noise in the data. I also question the appropriateness of using linear regression, which is based on bivariate correlation, on ordinally scaled variables. Given these concerns, I am not sure that their discussion follows from their data.
Response 1:
We sincerely appreciate the reviewer's comments and suggestions for our paper and addressed all issues raised below. We feel that the review process improved our manuscript substantially.
Comments 2:
The title states “preference for dogs and cats” yet you measured preference for dog and cat videos. You need to justify why the number of views and comments to dog and cat videos are reliable and valid measures of preference for dogs and cats. Perhaps the dog and cat videos are watched more than other types of videos because they are funnier than the other types of videos or cuter than other type of videos or more calming, etc.
Response 2:
We sincerely appreciate the reviewer for pointing out the issue of whether video views and comments are suitable as an indicator of preferences. We adjust terms such as "human preferences for dogs and cats" to "human preference for dog and cat videos" in related expressions in the new manuscript, and updated the new title “Human preferences for dog and cat in China: the current situation and influencing factors of watching online videos and pet ownership”.
Our hypothesis is that if a particular theme's videos receive higher views and comments, it likely indicates a relatively strong user preference for that theme. Although this approach has certain limitations, we believe that the high views and comments demonstrated through large-scale online data can reflect people’s preferences for different themes. In fact, Myrick’s article on YouTube cat video views indicates a significant correlation between people's affinity (preference) for cats and the views of cat-related online content [1]. Similarly, De Oliveira et al. found that most YouTube users (69.6%) actively seek and watch videos that align with their interests [2]. However, we confine our discussions strictly to online video preferences without extending beyond that scope. Additionally, we have included explanations in the introduction (see in lines 99-104).
1, Myrick, J.G. Emotion Regulation, Procrastination, and Watching Cat Videos Online: Who Watches Internet Cats, Why, and to What Effect? Comput. Hum. Behav. 2015, 52, 168–176, doi:10.1016/j.chb.2015.06.001.
2, De Oliveira, R.; Pentoney, C.; Pritchard-Berman, M. YouTube Needs: Understanding User’s Motivations to Watch Videos on Mobile Devices. In Proceedings of the Proceedings of the 20th International Conference on Human-Computer Interaction with Mobile Devices and Services; ACM: Barcelona Spain, September 3 2018; pp. 1–11.
Comments 3:
Lines 141-143: You state that questions 11 through 20 assess whether dogs and cats can partially fulfill the role of close family members. It is clear how questions 13, 16, and 20 can be used to answer that question. It is not clear how the other questions can be used to determine if dogs and cats can partially fulfill the role of close family members. Please explain.
Response 3:
We are grateful to the reviewer for highlighting the issue regarding the unclear purpose of other questions. The purpose of Questions 12, 14, and 15, as well as Questions 17 and 18, is to filter respondents who need to answer Questions 13, 16, and 20, respectively. We believe that responses from those respondents whose close family members are not nearby (e.g., not living together or absent) provide higher reference value for these questions. In the new manuscript, we have updated the description to clarify that the other questions were used to select respondents for answering questions 13, 16, and 20 (see in lines 148-152).
Comments 4:
Why are you using Pearson’s correlation and linear regression with your ordinally scaled data? Pearson’s r requires interval or ratio scaled data. Linear regression makes no sense with ordinally scaled (dummy coded) variables. Your results and discussion need to be rewritten to reflect a proper measure of association.
Response 4:
We sincerely appreciate the reviewer for pointing out the issue of ordinal variables being unsuitable for Pearson’s correlation and linear regression. We highly value this and have thus recalculated the parental correlation in dog and cat ownership using Spearman's rank correlation method [1], which is suitable for discrete ordinal variables. Our calculation shows that the Spearman's rank correlation coefficient is 0.43 (95% CI: 0.38 - 0.47), which is very close to the previous linear regression result (linear regression coefficient is 0.42, 95% CI: 0.34 - 0.51). We have updated the Method section (lines 192 and 199-200), Results section (lines 241–253) and the Discussion section (line 333 and 341) in the revised manuscript accordingly.
1. Zar, J.H. Spearman Rank Correlation. In Encyclopedia of Biostatistics; Armitage, P., Colton, T., Eds.; Wiley, 2005 ISBN 978-0-470-84907-1.
Comments 5:
Line 190: It does not make sense to take the mean of several Pearson’s rs as Pearson’s r is not linear. Consider using an r to z transform, taking the mean of the z-scores, and use a z to r transform.
Response 5:
We thank the reviewer for this suggestion to calculate the average using r-to-z transformation. In calculating the Spearman's rank correlation coefficients, we applied the r-to-z transformation to calculate the average and then performed the z-to-r inverse transformation to obtain the result. For the three Spearman's rank correlation coefficients (0.47 (95% CI: 0.38 - 0.55), 0.41 (95% CI: 0.33 - 0.50), and 0.40 (95% CI: 0.30 - 0.49)), the average obtained using the r-to-z transformation was 0.43 (95% CI: 0.38 - 0.47). In the Method section, we explained that the average was obtained through the r-to-z transformation in the new manuscript (see lines 198–199).
Comments 6:
Lines 191-192: Perhaps this is just a cultural difference. The first question on the questionnaire just asks about “grandparents” and not “paternal grandparents”. The second question on the questionnaire asks about “maternal grandparents”. How did you measure dog and cat ownership of paternal grandparents?
Response 6:
We are grateful to the reviewer for pointing out this issue. When writing the supplementary material, we inadvertently omitted the word "paternal" in " paternal grandparents" in Question1. We have now revised “grandparents” to ”paternal grandparents” in Question 1 of Table S2.
Comments 7:
Lines 220-226: To me, you are interpreting random noise in the data. That is, the means and medians do not look like they would be reliably different as they appear to be within each other’s IQRs. Do you have statistical results to support your statement that pet category is third or fourth most popular? If not, lines 289-290 in the discussion will need to be modified.
Response 7:
We sincerely appreciate the reviewer for pointing out the issue of random noise in the data. We have revised carefully the interpretation of our results in the Discussion section and updated the related wording to avoid overgeneralization in the new manuscript (see line 297). We have changed "humans prefer" to "humans may have a relative preference for".
Comments 8:
Lines 323-324: It could also represent an environmental basis – living with a pet while growing up may predispose a person to have a pet during adulthood. You need to explain how your data tease these two factors (nature vs nurture) apart.
Response 8:
We thank the reviewer for highlighting the issue of environmental basis. Due to the limitations of our existing data, we cannot clearly distinguish the impacts of genetic factors (nature) from environmental factors (nurture). However, we listed studies by Fall et al., which estimated the heritability of dog ownership using monozygotic and dizygotic twins, finding heritability estimates of 0.51 for males and 0.57 for females. We also referenced Jacobson et al., which found that genetic factors accounted for as much as 37% of the variance in pet play, also using twin data. These studies effectively tease apart the contributions of genetic and environmental factors to these behaviors. These studies indicate that dog ownership may have a genetic basis, and their estimates of heritability are very similar to our results (0.43, 95% CI: 0.38 - 0.47). Therefore, we believe that the similarity between these estimates provides strong support for the reliability of our conclusions. We believe that future research could better elucidate the relative contributions of these factors through more detailed designs, such as twin or adoption studies. We have added our discussion on environmental factors and potential future solutions at the end of this paragraph in the new manuscript (see lines 343–347).
Comments 9:
Lines 337-341: If I am understanding you correctly, you are reporting the percentage of people who answered “Yes” to question 6 through 10. You are giving the frequencies of each category but that is not the same as “relative importance”. More people might claim that they own a dog or cat for functional purposes (e.g. 100 people) than for emotional value (e.g. 90 people), but the functional purpose people may give an importance rating (which you did not measure) of 2 out of 7 while the emotional people may give an importance rating of 7 out of 7.
Response 9:
We are grateful to the reviewer for pointing out the issue regarding the term “relative importance”. To more accurately reflect our results, we have replaced “relative importance” with “selection frequencies” to clearly indicate that we are reporting the frequency of “yes” responses rather than an assessment of importance. We have also made corresponding revisions in the subsequent discussion in the new manuscript (see lines 350-352).
Comments 10:
Line 27: Insert a space between the hyphen and 0.51. As written, it looks like the CI runs to negative 0.51 which would imply that the CI contains 0.
Response 10:
We sincerely appreciate the reviewer for pointing out this, and we have made the necessary revisions in the new manuscript (see line 30).
Comments 11:
Line 50: “study” should be “studied”
Response 11:
We are grateful to the reviewer for highlighting this issue, and we have made the necessary revisions in the new manuscript (see line 53).
Comments 12:
Lines 111-112: Close the gap. The materials and methods section should not start on a separate page.
Response 12:
We thank the reviewer for bringing this to our attention, and we have made the necessary revisions in the new manuscript (see line 116-117).
Comments 13:
Line 149: Include the unit (e.g. years) with ages.
Response 13:
We appreciate the reviewer’s attention to this, and we have made the necessary revisions in the new manuscript (see line 157).
Comments 14:
Line 166: The sentence starting “Furthermore, we obtained…” does not belong in section 2.1.3 Douyin (TikTok) videos. Please put it in a separate section.
Response 14:
We value the reviewer’s insightful comment on this, and we have deleted this sentence in the new manuscript.
Comments 15:
Line 208: “datas et” should be “data set”
Response 15:
We sincerely thank the reviewer for noticing this, and we have made the necessary revisions in the new manuscript (see line 215).
Comments 16:
Figure 3: I recommend that you make the filled part of each bar lighter in color or make the value labels within each bar white. Either way you will increase the contrast of the value labels and bars making the graph more legible.
Response 16:
We are thankful for the reviewer’s insight into this, and we have lightened the filled portion of each bar in Figure 3 in the new manuscript (see line 263).
Comments 17:
Line 266: “Female” and “male” would be more appropriate than “Woman” and “man”.
Response 17:
We value the reviewer’s suggestion in this, and we have made the necessary revisions in the new manuscript (see line 273).
Comments 18:
Caption for figure 4: Indicate what the whiskers represent in the figures 4a-4c.
Response 18:
We are indebted to the reviewer for identifying this. The whiskers in Figures 4a-4c represent the standard error (SE) values. We have made updates in the new manuscript (see line 284).
Comments 19:
Line 291: “owning” should be “own” and “are” should be “is”
Response 19:
We commend the reviewer’s attention to this, and we have made the necessary revisions in the new manuscript (see line 298).
Comments 20:
Line 371: Lines 42-43 state 49% and 35% of households in the USA own dogs / cats. Line 371 states 33.4% and 22.5% of households own dogs / cats. I am guessing that the difference is due to the region (USA vs not specified). It would be nice if you made explicit the difference in these percentages.
Response 20:
We appreciate the reviewer’s careful observation. The data of 33.4% and 22.5% comes from Sydney, Australia. We have added the study region in the new manuscript (see line 384).
Comments 21:
Line 381: “data from Wiki” is not an appropriate citation or source.
Response 21:
We value the reviewer’s suggestion in this regard, and we have deleted this sentence in the new manuscript.
Comments 22:
Line 384: Consider deleting “real”. “Life” is “real” by definition.
Response 22:
We are thankful for the reviewer’s insight into this, and we have made the necessary revisions in the new manuscript (see line 395).
Comments 23:
Line 413: Consider “articles” instead of “research”.
Response 23:
We sincerely thank the reviewer for noticing this, and we have made the necessary revisions in the new manuscript (see line 425).

Reviewer 4 Report (New Reviewer)
Comments and Suggestions for Authors
Thank you for your paper, I found it interesting in engaging quantitative methods to explore human preferences for dogs and cats in China. Your paper outlines your research aim and question well and uses an excellent amount of relevant literature across disciplines to formulate and support your argument. I am not overly sure about your argument of preference for cats and dogs when talked about in terms of your video data, as to me it should be read as preference for cat and dog videos over other videos (e.g. cooking). I have some constructive comments for you to consider which I think will help you improve your paper as I am currently recommending major revisions.
Abstract and simple summary.
I think you could be clearer in that humans prefer cat and dog videos to other videos.
Introduction.
I would reword the first sentence as I don’t think it makes sense, what do you mean by dogs and cats being influential?
Does line 50-52 need a reference?
L90-95, I am not sure about the framing of this point in framing cats and dogs as an alternative form of entertainment.
I think your aims could be slightly clear when talking about preference, prefer from what, other types of videos, as a pet, as a form of entertainment?
Methods
This section was interesting and I had a few questions that came to my mind. To what extent do you account for overlap in your categorization of videos (Bilibili site). Could the same video occupy both the cat and dog categories, could the same video? When grouping cats, dogs, and cute pets, do you not lose the differences here? Did you do anything with the comments on Bilibili or was this just Douyin?
For your online questionnaire, I am unsure on the ethics of keeping the purpose of the questionnaire hidden. I think you need to say more about why you did this and what it achieves. How do you judge what is an ‘unreal’ answer? I see you questions often use ‘cat/dog’, did you survey at any point aim to gain different understandings of these animals in relation to owners?
2.2.3. – Did you leave room for other reasons beyond the 5 chosen for owning dogs and/or cats?
Results
3.3. I was wondering if people could chose multiple reasons for owning dogs and/or cats and how this is factored into your statistics? If people could only chose 1 of 5 options how do you account for multiple reasons i.e. both for emotional support and safety?
Is figure 3 better represented as a table?
I feel less qualified to comment on the statistical tests (e.g. the regression models).
Discussion
Again, I think there needs to be a greater recognition of people preferring cat/dog videos to gaming videos when talking about the online video data compared to the questionnaire. I think there is a nuance there that needs to be addressed rather than using the video data analysis to say humans prefer cats and dogs to other interests.
Conclusion
I think the conclusion could be expanded a little bit more and made a little bit clearer.
Misc
A lot of the manuscript is highlighted in yellow, I am not sure what this means, whether these were additions, or you just forgot to change this.
Author Response
Comments 1:
Thank you for your paper, I found it interesting in engaging quantitative methods to explore human preferences for dogs and cats in China. Your paper outlines your research aim and question well and uses an excellent amount of relevant literature across disciplines to formulate and support your argument. I am not overly sure about your argument of preference for cats and dogs when talked about in terms of your video data, as to me it should be read as preference for cat and dog videos over other videos (e.g. cooking). I have some constructive comments for you to consider which I think will help you improve your paper as I am currently recommending major revisions.
Response 1:
We thank the reviewer for their positive assessment of our manuscript and addressed all issues raised below. We believe that these comments and suggestions have greatly improved the quality of our work.
Comments 2:
I think you could be clearer in that humans prefer cat and dog videos to other videos.
Response 2:
We sincerely appreciate the reviewer for pointing out this issue. We have clearly stated in the Abstract and Simple Summary that humans relatively prefer dog and cat videos over other videos, which ranks among the top 3 out of 15 interests.
Comments 3:
I would reword the first sentence as I don’t think it makes sense, what do you mean by dogs and cats being influential?
Response 3:
We are grateful to the reviewer for highlighting this issue. We have revised the first sentence in the new manuscript (see lines 37-38). The term “influential” that we used previously actually refers to “important”.
Comments 4:
Does line 50-52 need a reference?
Response 4:
We thank the reviewer for bringing this issue to our attention. We have added the relevant URL for reference after this sentence in the new manuscript (see lines 56–57).
Comments 5:
L90-95, I am not sure about the framing of this point in framing cats and dogs as an alternative form of entertainment.
Response 5:
We sincerely appreciate the reviewer for pointing out this issue. We have removed the ambiguous term “forms of entertainment” in the new manuscript (see lines 98–99). Our intention was simply to compare human preferences for dogs and cats with those for other subjects.
Comments 6:
I think your aims could be slightly clear when talking about preference, prefer from what, other types of videos, as a pet, as a form of entertainment?
Response 6:
We are indebted to the reviewer for identifying this issue. We have added the relevant content later in the text. We compared preferences for dogs and cats with those for other hobbies and interests, such as music and games (see lines 98–99).
Comments 7:
This section was interesting and I had a few questions that came to my mind. To what extent do you account for overlap in your categorization of videos (Bilibili site). Could the same video occupy both the cat and dog categories, could the same video? When grouping cats, dogs, and cute pets, do you not lose the differences here? Did you do anything with the comments on Bilibili or was this just Douyin?
Response 7:
We are grateful to the reviewer for highlighting these issues. We will address each point in turn.
Regarding the overlap in video categorization:
Videos on Bilibili channels may indeed overlap across categories. Video grouping is determined by both Bilibili’s official system and the video uploaders themselves. Uploaders select tags that best fit their video theme to reach a broader audience interested in that category, and Bilibili then organizes channels based on these tags. As a result, a single video may appear in multiple channels. We excluded all overlapping instances and recalculated the results, which remained similar to the previous findings. We find that current human preferences for dog and cat videos are relatively higher than for most other interests, ranking among the top 3 out of 15 interests. We have made some updates in Methods (see lines 182-183), the results (see lines in 226-239) and discussion section.
Regarding cats, dogs, and cute pets grouping:
When analyzing Bilibili data, we treated dogs and cats as a whole category. In this part of the study, we did not aim to examine subtle differences between dog and cat videos. Our goal was simply to compare human preferences for pets versus other categories, such as music or game.
Regarding comments:
We only sampled comments from Douyin to estimate overall user demographics and analyze the factors influencing human preferences for dogs and cats. Douyin’s broader and more representative user base made it ideal for this purpose. We did not conduct a similar analysis for Bilibili due to the data availability limitation for the data collection platform.
Comments 8:
For your online questionnaire, I am unsure on the ethics of keeping the purpose of the questionnaire hidden. I think you need to say more about why you did this and what it achieves. How do you judge what is an ‘unreal’ answer? I see you questions often use ‘cat/dog’, did you survey at any point aim to gain different understandings of these animals in relation to owners?
Response 8:
We sincerely appreciate the reviewer for highlighting these issues. We will address each point in turn.
Regarding the concealed purpose:
The sentence "The purpose of the questionnaire was not disclosed to the participants." refers to withholding the conclusions we aim to verify. Directly disclosing our conclusions in the questionnaire could potentially bias participants' responses. This approach aligns with standard research practices [1]. In the questionnaire, we only informed participants that the questionnaire is used for scientific study and ensured proper handling of anonymized data. To avoid ambiguity, we have revised and updated the related statements in the new manuscript (see lines 141–144).
- Li, Y.; Wan, Y.; Zhang, Y.; Gong, Z.; Li, Z. Understanding How Free-Ranging Cats Interact with Humans: A Case Study in China with Management Implications. Biol. Conserv. 2020, 249, 108690, doi:10.1016/j.biocon.2020.108690.
Regarding unreal answer:
Unreal answers refer to questionnaires where participants selected the first option for all questions or provided clearly erroneous answers.
Regarding dogs/cats:
We consider dog and cat ownership to be equivalent levels of pet-keeping behavior; therefore, both options are presented alongside each other in the questionnaire. In our survey, we treat dog and cat ownership as a whole category rather than distinguishing between different relationships owners may have with dogs and cats.
Comments 9:
2.2.3. – Did you leave room for other reasons beyond the 5 chosen for owning dogs and/or cats?
Response 9:
We sincerely appreciate the reviewer’s attention to this issue. Our aim is to explore the most common reasons why people own dogs or cats. Therefore, in this study, we did not provide participants with options beyond the five predefined reasons for dog and/or cat ownership.
Comments 10:
I was wondering if people could chose multiple reasons for owning dogs and/or cats and how this is factored into your statistics? If people could only chose 1 of 5 options how do you account for multiple reasons i.e. both for emotional support and safety?
Response 10:
We thank the reviewer for bringing these issues to our attention. Each respondent who owned or will own dogs or cats answered all five questions regarding the reasons for pet ownership, with response options of “yes” or “no”, and each question required response. We calculated the proportion of respondents selecting “yes” or “no” for each question to identify the most common reasons why people choose to own dogs or cats.
Comments 11:
Is figure 3 better represented as a table?
Response 11:
We value the reviewer’s suggestion in this. We have added a new table after Figure 3 to show the selection frequency for each of the five questions in the new manuscript (see lines 269-270).
Comments 12:
I feel less qualified to comment on the statistical tests (e.g. the regression models).
Response 12:
We sincerely appreciate the reviewer's attention to the statistical tests. We have adopted more appropriate statistical tests for our study to ensure the reliability and validity of the results. Using regression analysis is a classical way to calculate heritability in quantitative genetics. We have calculated the parental correlation in dog and cat ownership using Spearman's rank correlation method [1], which is suitable for discrete ordinal variables. Our calculation shows that the Spearman's rank correlation coefficient is 0.43 (95% CI: 0.38 - 0.47), which is very close to the previous studies by Fall et al., which estimated the heritability of dog ownership using monozygotic and dizygotic twins (0.51 for males and 0.57 for females), and by Jacobson et al., which found that genetic factors accounted for as much as 37% of the variance in pet play. These findings provide a solid basis for the reliability of our conclusions. Furthermore, we used a typical quantitative genetics method (i.e., Animal Models that incorporated the pedigree information using the MCMCglmm package in R) to recalculate the potential heritability, and the resulting estimate was 0.35 (95% CI: 0.29-0.39). This value is very close to our Spearman's rank correlation coefficient results. These similar estimates of potential heritability obtained from different methods indicate that there may indeed be a heritable component to dog and cat ownership behavior.
- Zar, J.H. Spearman Rank Correlation. In Encyclopedia of Biostatistics; Armitage, P., Colton, T., Eds.; Wiley, 2005 ISBN 978-0-470-84907-1.
Comments 13:
Again, I think there needs to be a greater recognition of people preferring cat/dog videos to gaming videos when talking about the online video data compared to the questionnaire. I think there is a nuance there that needs to be addressed rather than using the video data analysis to say humans prefer cats and dogs to other interests.
Response 13:
We are grateful to the reviewer for highlighting the need to emphasize preferring cat/dog videos to other videos. We understand and value this insight. We have recognized the necessity to more clearly articulate that people have a stronger preference for dog and cat videos compared to other types, such as music and game videos. Therefore, we have revised the term “human preferences for dogs and cats” in the related statements to “human preferences for dog and cat videos” in the new manuscript.
Comments 14:
I think the conclusion could be expanded a little bit more and made a little bit clearer.
Response 14:
We are thankful for the reviewer’s insight into this. We have appropriately expanded the conclusion to further clarify the main findings of the study and emphasize the potential implications of these findings for future research and practice in the new manuscript (see lines 465-478).
Comments 15:
A lot of the manuscript is highlighted in yellow, I am not sure what this means, whether these were additions, or you just forgot to change this.
Response 15:
We appreciate the reviewer’s careful observation. The highlighted sections in our manuscript were used to respond to comments from other reviewers in the previous round. All previous highlights have now been removed, and new highlights have been added based on the comments and suggestions from this round of review in the new manuscript.

Round 2
Reviewer 3 Report (New Reviewer)
Comments and Suggestions for Authors
Review of animals-3261832.
I thank the authors for addressing my issues. Below are some minor issues that you may want to address:
In table 1, three decimal places are sufficient.
Figure 3 and table 2 are redundant. I recommend removing one.
In section 3.4, you should include the degrees of freedom with the t-tests. That is, t(df) = 3.7, … where df is the degrees of freedom for the tests
Line 323: Delete “that” in “Since that the results here…”
Paragraph starting at 331: I am not convinced that your results are similar to the heritability results. Heritability is a measure of the proportion of variance explained by genetics. You would need to square the correlation coefficient to get the proportion of variability that is explain in one variable by the other variable. Shouldn’t you be comparing 0.43² = 0.185 to 0.51 and not 0.43 to 0.51.
Author Response
Comments 1:
I thank the authors for addressing my issues. Below are some minor issues that you may want to address:
Response 1:
We appreciate the reviewer’s positive assessment of our manuscript, and we have addressed all the issues raised. We are confident that these comments and suggestions have significantly enhanced the quality of our work.
Comments 2:
In table 1, three decimal places are sufficient.
Response 2:
We are grateful to the reviewer for identifying this issue. We have now revised the figures to retain three decimal places in the revised manuscript, as shown in Table 1 (lines 252-253).
Comments 3:
Figure 3 and table 2 are redundant. I recommend removing one.
Response 3:
We sincerely appreciate the reviewer’s pointing out of this issue. We have removed Table 2 from the revised manuscript.
Comments 4:
In section 3.4, you should include the degrees of freedom with the t-tests. That is, t(df) = 3.7, … where df is the degrees of freedom for the tests
Response 4:
We sincerely appreciate the reviewer’s highlighting of this issue. We have added the degrees of freedom (df) values at the corresponding positions, as follows: t = 3.7, P < 0.001, df = 1004 in line 271; t = 137.4, P < 0.001, df = 119672 in line 272; t = 7.6, P < 0.001, df = 23 in line 274.
Comments 5:
Line 323: Delete “that” in “Since that the results here…”
Response 5:
We appreciate the reviewer’s careful observation. We have removed the word “that” from the revised manuscript (see line 321).
Comments 6:
Paragraph starting at 331: I am not convinced that your results are similar to the heritability results. Heritability is a measure of the proportion of variance explained by genetics. You would need to square the correlation coefficient to get the proportion of variability that is explain in one variable by the other variable. Shouldn’t you be comparing 0.43² = 0.185 to 0.51 and not 0.43 to 0.51.
Response 6:
We are grateful to the reviewer for highlighting this issue. There was indeed some misunderstanding in this part. The Spearman rank correlation coefficient (0.43) was used to assess the correlation between the pet ownership behaviors of offspring and their parents. This regression analysis is a classical way to calculate heritability in quantitative genetics before the more advanced statistical methods developed (e.g., ‘Animal Models’). On the other hand, we also used MCMCglmm package (i.e., a typical quantitative genetics method-Animal Models-of heritability estimating, which incorporated pedigree information) to estimate the potential heritability of dog and cat ownership behavior. (methods: lines 195-202; results: lines 241-247). This yields a value of 0.35. This heritability value is very close to the results reported by Jacobson (0.37) and lower than those reported by Fall (0.51 for men and 0.57 for women). We have revised the description in lines 331-336 of the new manuscript, clearly stating that our heritability estimate is 0.35 and comparing it to the values of 0.37 and 0.51.

This manuscript is a resubmission of an earlier submission. The following is a list of the peer review reports and author responses from that submission.
Round 1
Reviewer 1 Report
Comments and Suggestions for Authors
The paper addresses an interesting and relevant topic, exploring human preferences for dogs and cats using contemporary data from online platforms like Bilibili and Douyin. This approach provides original insights into pet ownership patterns and preferences.
One of the advantages of such procedure, is that the use of large datasets from popular online platforms and questionnaires provides a large dataset for the analysis.
However, the paper could benefit from a stronger theoretical foundation. While it provides descriptive statistics and some regression analysis, there is a lack of deeper theoretical engagement with why these preferences exist. The paper could be improved by integrating relevant theories from psychology, sociology, or animal behavior more explicitly to contextualize the findings.
The study heavily relies on data from online platforms (Bilibili and Douyin), which may introduce significant biases. The demographic that engages with these platforms might not be representative of the general population, particularly older individuals or those without access to technology. This limitation should be more critically addressed in the discussion.
The design and implementation of the questionnaire could be more thoroughly explained. It is not clear how representative the sample is or how well the questions capture the nuances of human-pet relationships. Additionally, the questionnaire might suffer from self-report biases, which could distort the results.
While the paper provides statistical analysis, the interpretation of these results is somewhat superficial. The discussion often reiterates the findings without offering deeper insights or implications. There is an opportunity to explore the broader significance of these preferences, perhaps linking them to trends in urbanization, cultural shifts, or economic factors more critically.
The paper attempts to generalize findings from a specific cultural context (China) to broader human behavior. While the data is valuable, the authors should be cautious about overgeneralizing these results without considering cultural, economic, and social differences across regions.
The paper’s structure could be improved for clarity. Some sections are dense and difficult to follow, particularly where the methodology and results are concerned.
The supplementary materials are referenced but not thoroughly integrated into the main text. It would be helpful to include a more detailed discussion of these materials within the paper itself, to help the study’s overall narrative and comprehension.
Overall, the paper presents an interesting study with large data, but the quality of the data is not discussed while it is a major bias and could impact most of the claims made by the authors.
Reviewer 2 Report
Comments and Suggestions for Authors
This is an interesting paper and the use of Chinese social media is novel. My primary concern relates to the linear regression analysis. I may be misunderstanding how this was done but it seems like the dependent variable (whether family members have had dogs/cats) is not ordinal. If one is owning dogs and two is owning cats this is a nominal not ordinal category. If this is not the case this needs to be clarified. The presentation of these data in graph form is hard to read -- can we see the actual regression results?
It is likely the case that the sample is biased toward younger and better off users but this is noted as a limitation so that is fine.
Comments on the Quality of English LanguageThere are some language issues -- not horrible but it will need a good bit of editing. The reporting of the data is better than the literature review.
Reviewer 3 Report
Comments and Suggestions for Authors
The paper is very well written and touches a very interesting issue, by opening a stimulating discussion on the influence of social media on people preferences and behaviors. In today's digital age, consumers are constantly exposed to content that aligns with their interests. In particular, short videos are meant to capture the attention of especially younger audiences, showing products and brands that align with their preferences. However, the platform's viral nature can induce them to make impulsive decisions. As a matter of fact, Tick Tock has emerged as a game-changer in the world of social media and has a profound impact on consumer behavior. However, as an animal welfare expert, I don’t feel confident to express a qualified opinion about the relevance of this paper and the significance of the results. Hence, I would suggest to also involve other reviewers, namely experts on social media and digital communication.
In the same vein, I would be prudent to conclude that human preferences for dogs and cats may also be influenced by genetic variations on the basis of this study. Actually, I am afraid it would be not sufficient to ask if grandparents and parents have or have ever had a dog/cat to affirm that genetic variations may affect humans preferences for dogs and cats. At line 266 authors make reference to a large twin study including 35,035 twin pairs.
In essence, I believe that the open question about whether the human interest in dogs and cats has low or high heritabily deserves further consideration and should be approached by using quantitative behavioral genetics methodologies. However, this is outside my field of expertise.